# Does Internet Use Aggravate Public Distrust of Doctors? Evidence from China

Lingpeng Meng [1] , Xiang Yu [2], Chuanfeng Han [2,3] and Pihui Liu [3,*]

1 China Institute of FTZ Supply Chain, Shanghai Maritime University, Room 427, Container Supply Chain Building, Harbour Avenue No. 1550, Pudong District, Shanghai 201306, China; lpmeng@shmtu.edu.cn
2 School of Economics and Management, Tongji University, 1239 Siping Road, Shanghai 200092, China; yux@tongji.edu.cn (X.Y.); hancf@tongji.edu.cn (C.H.)
3 School of Management Engineering, Shandong Jianzhu University, Jinan 250000, China
* Correspondence: lph@tongji.edu.cn

**Abstract:** The internet has exacerbated the spillover of medical information, and changes in the quantity, quality, and scope of information supply also affect public trust in doctors, which is of great significance to the construction of a harmonious physician–patient relationship. The objective of this study is to explore the relationship between internet use and residents' trust in doctors using data from the China Family Panel Studies for 2018. The empirical investigation utilizes an endogenous switching regression model (ESR) to overcome the endogeneity bias. Our results indicate that internet use is negatively associated with residents' trust in doctors ($\beta = 0.07$, $p < 0.05$). Specifically, the patient trust of internet users is nearly 7 percent less than that of non-internet users. Nevertheless, residents with higher dependence on traditional media such as television, newspapers, magazines and radio as an information channel show stronger patient trust. Finally, the results of the subsample analysis indicate a need to focus on older and less-educated residents, who are more vulnerable and more likely to be affected.

**Keywords:** internet; traditional media; patient trust; endogenous switching regression model; China; CFPS 2018

## 1. Introduction

China has faced a serious problem of violence against doctors (also known as "medical disturbance") in recent years. According to a survey of more than 2000 medical staff across China in 2020, more than 80% have experienced violence from patients or patients' families in hospitals or their offices [1]. The types of violence include verbal violence, such as abuse and threats, and physical violence with the hands or even instruments. These vicious events may have effects that extend far more beyond the doctor–patient relationship. For instance, research has contended that workplace violence may increase turnover intention among nurses and decrease student willingness to choose being a doctor as a career [2].

The roto cause of patients' frequent violence toward doctors is related to patients' lack of trust in doctors [3]. Patients' trust in doctors represents how confident a patient is that doctors will act in the patient's best interest and supply suitable therapy [4]. Trust is the foundation of a harmonious relationship between patients and physicians, and a lack of patient trust can lead to serious consequences. On the one hand, patients who are skeptical of physicians are significantly less likely to communicate their health conditions in detail with health care providers, and to comply with the doctor's proposed instruction, thereby limiting the positive effects of medical treatment [5]. On the other hand, a reduction in patient trust may also hurt doctors' feelings. Mistrusting patients can reduce the enthusiasm of doctors to work and further increase the possibility of medical disputes. Moreover, a lack of trust in physicians is associated with patient violence [6,7].

The crisis of trust in doctors is becoming a global problem. In a report on pooled data on public trust in doctors and medical leaders, a survey from 29 industrialized countries indicated that public trust in the leaders of the medical profession has declined sharply [8]. The situation may be even worse for China, the world's largest developing country. A study on China shows that from 2011 to 2016, patients' trust in doctors decreased significantly, and nearly 20% have completely lost their trust in medical service providers [9]. In fact, Chinese people's trust in doctors has been declining slowly for a long time, but the trend has been accelerating recently. When the People's Republic of China was born in 1949, all medical institutions became publicly owned. At that time, the relationship between the patient and physicians was quite simple, without considering many benefits individually. After reform and opening in 1984, the Chinese medical system also changed with medical free market reforms. The government's investment in medical service institutions has decreased, and a hospital's management mainly relies on its own income. Physicians tend to provide treatment plans based on their own profit instead of considering patients. This leads to the increase in patients' distrust of doctors and the aggravation of the contradiction between doctors and patients. Although China has since led the reform of the diversified medical service supply model, the problem of lost patient trust remains unresolved [10]. Only by understanding the drivers of and barriers to the transition of public trust in doctors can more effective policies be developed to promote the improvement of trust in doctors.

Numerous studies have examined the determinants of trust in doctors. The various factors include demographic characteristics, psychological factors, social circumstances, and government administrative interventions [11–14]. However, although information and communication technology (ICT) has played an essential role in modern life, its potential impact on public trust in doctors has not yet been studied. The internet is the most representative ICT in today's digital society [15–18]. According to the Internet World Stats, nearly 2/3 of the global population were internet users by the end of July 2021. Like many other countries, China has seen a substantial increase in the number of internet users. According to the newest report from the China Internet Network Information Center (CNNIC), there were 989 million internet users in China as of December 2020.

The rapid popularization of the internet use has brought about substantial changes in individual behaviors, living habits and values. The internet has become a leading source for the general public and patients to search for health information, communicate, and make decisions, which may change their attitude toward physicians [19,20]. On the one hand, individuals may increase their trust in doctors by browsing health information, communicating with doctors online, and accessing some primary health care services. However, negativity bias theory emphasizes that negative news is more attractive to people than positive news [21,22]. Therefore, netizens might browse negative news related to "medical disturbances", thereby reducing their trust in physicians. For example, in 2019, a deputy chief physician of the Emergency Department of Beijing Civil Aviation General Hospital died of a malignant injury caused by a patient's family; this case was widely distributed on the internet and triggered strong reactions and continued discussion. Since information on the internet can be mixed in quality, it is important to understand how the internet can affect patient trust, which is crucial to improving physician–patient relationships. However, the limited literature does not supply direct answers regarding the effect of internet use on patient trust.

Therefore, this paper poses the question of whether internet use could affect public trust in physicians in China. To investigate whether there is a plausibly causal link between these variables, we used a survey from the 2018 China Family Panel Studies (CFPS) and multiple econometric methods. Compared with existing studies, the marginal contributions of our research are as follows. First, to the best of our knowledge, this is the first study, from the perspective of information technology, to explore the effect of internet use on patient trust in an attempt to understand the strained relationship between physicians and patients. Although the impact of the internet on the economy and society has been empirically studied, much less is known about its impact in terms of physician–patient relations.

Second, endogenous switching regression models are employed to address potential self-selection bias, which provides a change in the level of patient trust. Moreover, we consider how different people hold heterogeneous attitudes toward information channels and have disparate behaviors. Furthermore, subsample analysis points to the heterogeneity of these effects across different demographic groups.

The remainder of this article is organized as follows. Section 2 presents the study background and literature review. Section 3 presents the research methods, including sampling and data sources and the analytical procedures, and the penultimate section describes and discusses the findings. Finally, Section 5 concludes the article.

## 2. Literature Review

### 2.1. Research on Patient Trust

Patient trust, also known as public trust in physicians or trust in health care providers, reflects individual confidence that health care professionals will act in the patient's best interest and provide appropriate health care [4,23]. Although prior investigations have implemented diverse approaches to measure patient trust, several scholars have agreed that patient trust can also be evaluated by the extent to which residents perceive that their care provider is credible, or their willingness to believe the information provided by health care professionals [24].

In general, patient trust is the result of the interaction of patients and physicians, which often depends on the characteristics of the medical care provider and the social circumstance under which they could be influenced by the government [11,12]. Scholars tend to group the influence of physicians into several categories, such as technical competence, interpersonal, predisposing factors, and structural/staffing factors [24]. Research has demonstrated that physicians' technical competence, treating patients sincerely, and patient-centered behavior can enhance patient trust [11]. Some studies consider the effect of communication between patients and physicians on trust in health care providers. A study in Israel has demonstrated the importance of communication about the enhancement of patient-perceived control over their own health [25–27].

Furthermore, the characteristics of the patient also have considerable effects. For example, a cross-sectional study suggests that higher stress and lower self-rated health are linked to decreasing trust in medical care providers [10]. In addition, demographic characteristics, such as gender, education, and age, lead to different trust levels toward physicians [9]. Additionally, existing studies indicate that the government's administrative interventions effectively shape public opinion toward physicians, and may enhance patient trust [13]. However, few studies have considered how internet use affects patient trust. Although the research results indicate that patient trust can play an important role in compliance with medical care providers and improve health outcomes, patient trust is still not fully understood and requires further exploration [28,29].

### 2.2. Internet Use and Patient Trust

The internet has exacerbated the spillover of medical information, and changes in the quantity, quality, and scope of information supply also affect patient trust. There is indirect evidence that may help to explain how patient trust is affected by internet use. On the one hand, the internet serves as an effective channel for health information and may reduce knowledge asymmetry between patients and health care providers [30,31]. Li et al. found that internet use has lowered the obstacles to learning about common ailments, resulting in a substitution impact of self-treatment for hospital care [32]. Chang et al. found that the use of health information on the internet is an effective way to improve the electronic health literacy of the elderly [33]. Due to the internet's flexibility and real-time interaction possibilities, it facilitates health community mobilization and physician–patient online communication [34–36]. This feature enables patients to send and receive text messages to obtain health information and request health care providers' real-time guidance, which also enables patient–physician online interaction and thus upgrades patient trust [37].

On the other hand, convenient access to health information also means a higher probability of being exposed to negative news and reports, which can result in higher anxiety and lower personal trust. Since people tend to be attracted by negative news more easily than by positive news, to gain more attention, internet media prefer to report news with the best-selling topics, which may damage patients' perceptions of doctors [38–40]. Misreports about the national medical system and doctors can give rise to the impression that national medical problems are severe, and physicians do not deserve to be trusted. Liu et al. found that information regarding insurance on the internet is usually negative and, thus, has a deterrent effect on insurance participation [41].

Moreover, health information on the internet can be misleading: a recent study evaluating more than 10 thousand websites shows that online health information is far from reliable [14]. Additionally, more frequent online health research is often associated with higher health anxiety [42]. However, when inconsistency appears between patient self-diagnosis expectations and their doctor's diagnosis, online information reduces trust in physicians, especially for those spending more time on the internet [5].

## 3. Materials and Methods

### 3.1. Source of Data

The data used in this study were from the China Family Panel Studies (CFPS), which is a representative annual longitudinal survey of the Chinese population. CFPS collected extensive information in 25 provinces in China (excluding Xinjiang, Qinghai, Tibet, Ningxia, Inner Mongolia and Hainan). A multistage probability strategy was used in this nationwide survey to increase efficiency and represent 95% of the composition of the Chinese population. Based on computer-aided investigation technology, data were collected through face-to-face interviews by experienced statisticians trained specifically for the CFPS program to ensure the high quality of the data. The database is public, and a more detailed description of the sampling design and process can be obtained from its website (http://www.isss.pku.edu.cn/cfps/index.htm, accessed on 12 October 2021). CFPS focuses on the economic and non-economic well-being of the Chinese population, as well as a wide range of information including economic activity, educational outcomes, family relationships and family dynamics, population migration, and health, providing a high-quality database for academic research and public policy analysis [43]. Since 2010, CFPS has conducted a survey every two years. In each survey wave, about 42,000 people living in 15,000 households were surveyed using face-to-face computer-assisted personal interviews. Six waves have been conducted. Considering the dynamic changes in residents in China, we use the 2018 wave sample. In 2018, the number of internet users in China had exceeded 800 million, the largest in the world. Meanwhile, the internet had become the main channel for Chinese residents to obtain information on doctor–patient disputes and medical matters, which was not significantly different from today. Therefore, the results obtained using the data in 2018 are universal. The CFPS 2018 sample consists of a total of 33,326 individuals. The cross-sectional response rate of the sample was 67.4%. We eliminated individuals who had missing values, declined to answer or replied with "unclear" and "difficult to say". As a result, 26,113 observations are used for analysis: 13,962 internet users and 12,151 non-internet users.

### 3.2. Variable Measurement

Following Kim and Choi, 2017 and Siaw et al., 2020 [44,45], we use a binary indicator equal to one if an individual uses at least mobile internet through any device (e.g., computer, smartphone, and iPad) in his or her daily life, and zero otherwise. Patient trust is measured by the question: "What is the level of your trust towards Americans?" Respondents were asked to select a number from 0 to 10. The number 0 was the lowest score and meant "very distrustful", while 10 was the highest score that meant "very trusting".

In addition, following the existing literature [46,47], we control for a range of demographic characteristics and covariates that may be correlated with internet use and patient

trust. Specifically, the demographic categories cover the personal characteristics of the respondent, such as gender, age, education, insurance status, personal income, marital status and whether he or she is a non-urban resident. The attitude categories consist of general trust intention and life satisfaction. The definitions and descriptive statistics of the major variables are presented in Table 1.

**Table 1.** Definitions of variables.

| Variables | Description | Mean | Std. Dev. | 95%CI |
|---|---|---|---|---|
| Dependent Variable | | | | |
| Patient Trust | Patient trust in physicians | 6.782 | 2.207 | [6.755, 6.809] |
| Explanatory Variables | | | | |
| Internet Use | Internet use status (1 = internet users, 0 otherwise) | 0.535 | 0.499 | [0.529, 0.541] |
| Urban | Urban respondent (1 = yes, 0 otherwise) | 0.508 | 0.500 | [0.502, 0.514] |
| Gender | Sex of respondent (1 = male, 0 otherwise) | 0.490 | 0.500 | [0.483, 0.496] |
| Age | Age of respondent (years) | 46.859 | 15.127 | [46.67, 47.04] |
| Education | Education level of respondent (1 = uneducated, 2 = primary education, 3 = secondary education, 4 = tertiary education) | 2.496 | 0.979 | [2.482, 2.508] |
| Marital Status | Marital status of respondent (1 = married, 0 otherwise) | 0.837 | 0.370 | [0.832, 0.841] |
| Income | Natural logarithm of annual personal income | 11.042 | 0.987 | [11.03, 11.05] |
| Insurance | Medical insurance status (1 = yes, 0 otherwise) | 0.924 | 0.266 | [0.920, 0.926] |
| Trust Intention | Whether respondent prefers to trust others (1 = yes, 0 otherwise) | 0.556 | 0.497 | [0.550, 0.562] |
| Life Satisfaction | Satisfaction with current life (1–5) | 3.996 | 0.950 | [3.984, 4.007] |
| Health condition | Self-rated health (1 = excellent, 2 = very good, 3 = good, 4 = fair, 5 = poor) | 2.952 | 1.221 | [2.943, 3.002] |
| Mobile Ownership | Mobile ownership status (1 = mobile owner, 0 otherwise) | 0.936 | 0.245 | [0.933, 0.939] |

Note: Std. dev. is standard deviation. CI is confidence intervals.

### 3.3. Analytic Strategy

The process of empirical analysis in this study is shown in Figure 1.

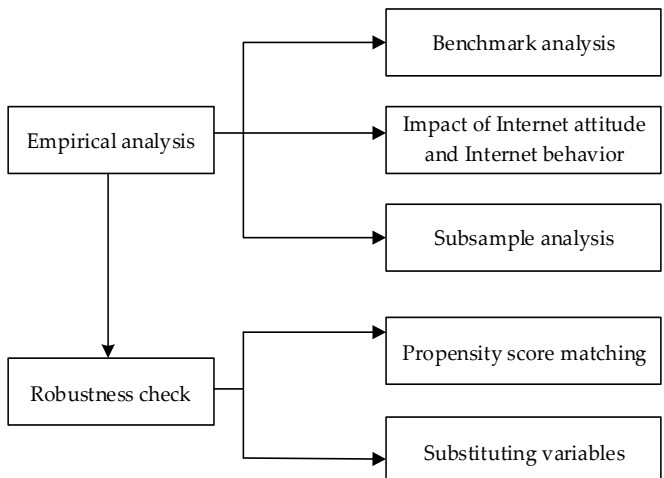

**Figure 1.** The empirical analysis process.

The Benchmark regression model for patient trust and internet use takes the following form:

$$Patient\_trust_i = \alpha_0 + \beta_0 Internet_i + \gamma X_i + province_i + \varepsilon_i \tag{1}$$

where *Patient_trust_i* represents the outcome of trust in physicians for individual *i*. *Internet_i* is a binary internet use indicator and is independent and identically distributed. $X_i$ is a vector of control variables that includes demographic characteristics such as gender, age, education, income, and marital status. *province_i* represents province fixed effects. $\alpha_0$ is a

constant term; $\beta_0$ and $\gamma$ are the parameters to be estimated; and $\varepsilon_i$ is an error term. OLS and ordered probit estimation are used as benchmark models.

However, selection bias occurs when the decision a resident makes whether to use the internet depends on observable and unobservable covariates simultaneously. For example, residents with higher income are more likely to have access to the internet, while other unobservable factors may also play a role in internet usage and patient trust.

To overcome this problem, we use endogenous switching regression (ESR) with the maximum likelihood algorithm and model the whole process in two stages: resident decisions to use the internet and the impact of internet usage on patient trust [48,49]. Therefore, we define the selection function as follows (stage 1):

$$Internet^*{}_i = S_i\alpha + \eta_i \; with \; Internet^*{}_i = \begin{cases} 1 \; if \; Internet^*{}_i > 0 \\ 0 \; otherwise \end{cases} \tag{2}$$

where $Internet^*{}_i$ represents latent internet usage utility. If the utility gain from using the internet is greater than that from not using the internet, Internet = 1; conversely, Internet = 0. We include $S_i$, which contains endogenous variables, including the variable "smartphone ownership", following [50].

Then, we turn to stage 2 and obtain two regression equations that provide outcomes under different conditions:

$$Y_{1i} = X_{1i}\beta_1 + \varepsilon_{1i} \; if \; Internet_i = 1 \tag{3}$$

$$Y_{2i} = X_{2i}\beta_2 + \varepsilon_{2i} \; if \; Internet_i = 0 \tag{4}$$

where $X_{1i}$ and $X_{2i}$ are consistent with Equation (1). Error terms $\eta_i$, $\varepsilon_{1i}$ and $\varepsilon_{2i}$ are assumed to follow a trivariate normal distribution with a zero mean vector. On the basis of the above equations, we can express the expected patient trust outcome of people who use or do not use the internet as Equations (5) and (6). In addition, the expected counterfactual with different internet usage habits can be defined as in Equations (7) and (8).

$$E(Y_{1i}|Internet_i = 1) = X_{1i}\beta_1 + \delta_{1\eta}\lambda_{1i} \tag{5}$$

$$E(Y_{2i}|Internet_i = 0) = X_{2i}\beta_2 + \delta_{2\eta}\lambda_{2i} \tag{6}$$

$$E(Y_{2i}|Internet_i = 1) = X_{2i}\beta_2 + \delta_{2\eta}\lambda_{1i} \tag{7}$$

$$E(Y_{1i}|Internet_i = 0) = X_{1i}\beta_1 + \delta_{1\eta}\lambda_{2i} \tag{8}$$

For residents who decide to use the internet, the unbiased average treatment effect on the treated (ATT) patient's trust can be calculated as the difference between (5) and (6).

$$ATT = E(Y_{1i}|Internet_i = 1) - E(Y_{2i}|Internet_i = 0) \tag{9}$$

## 4. Results and Discussion

### 4.1. Descriptive Results

Data on demographic and socioeconomic characteristics and patient trust collected using survey questionnaires are described and discussed in this section. Individuals are classified into treatment (internet use) and comparison (non-internet use) groups (see Table 2). The results show that 13,962 (53%) of the respondents are internet users, while 12,151 (47%) of the respondents are non-internet users. Internet users and non-internet users appear to show significant differences across the main covariates. For example, younger respondents are more likely to be internet users. The average age of internet users is approximately 47 years, whereas that of non-internet users is 57 years. The trust intention dummies reveal that internet users are more willing to trust strangers than non-internet users are, which significantly differs from the patient trust status. Urban residents are also more likely to be internet users.

**Table 2.** Description of the variables.

| Variables | Internet Users N = 13,962 | | Non-Internet Users N = 12,151 | | Mean Difference |
|---|---|---|---|---|---|
| | **Mean** | **Std. Dev.** | **Mean** | **Std. Dev.** | |
| Dependent Variable | | | | | |
| Patient Trust | 6.653 | 2.102 | 6.931 | 2.313 | 0.278 *** |
| Explanatory Variables | | | | | |
| Urban | 0.600 | 0.490 | 0.401 | 0.490 | −0.199 *** |
| Gender | 0.512 | 0.500 | 0.466 | 0.499 | −0.046 *** |
| Age | 38.368 | 12.359 | 56.615 | 11.764 | 18.247 *** |
| Education | 2.987 | 0.778 | 1.933 | 0.879 | −1.054 *** |
| Marital Status | 0.798 | 0.402 | 0.881 | 0.324 | 0.084 *** |
| Income | 11.334 | 0.840 | 10.707 | 1.035 | −0.627 *** |
| Insurance | 0.918 | 0.274 | 0.930 | 0.255 | 0.012 *** |
| Trust Intention | 0.595 | 0.491 | 0.512 | 0.500 | −0.083 *** |
| Life Satisfaction | 3.899 | 0.903 | 4.107 | 0.989 | 0.208 *** |
| Health Condition | 2.713 | 1.086 | 3.227 | 1.307 | 0.513 *** |
| Mobile Ownership | 0.998 | 0.039 | 0.864 | 0.342 | −0.134 *** |

Note. * $p < 0.1$, ** $p < 0.05$, *** $p < 0.01$. Std. dev. is standard deviation.

Table 2 also contains a summary of the outcome variables, where non-internet users appear to have higher patient trust than internet users. However, since the confounding factors should be controlled, these results cannot be used for inference regarding the impact of Internet use on an individual's patient trust. Hence, further analyses are conducted to obtain more reliable results.

*4.2. Benchmark Equation Results*

Table 3 presents the results for individual trust in physicians. Columns (1) and (4) show a simple regression of internet use on patient trust. The estimate is negative and statistically significant, implying that an individual that uses the internet tends to have, on average, lower patient trust. Columns (2) and (5) include individual characteristics to control for factors that affect patient trust and covary with individual latitude. The similar negative coefficient indicates that for a given individual, internet use, on average, lowers patient trust.

Columns (3) and (6) display our benchmark specification, as in columns (1) and (2), including province fixed effects. The estimate is again statistically significant ($\beta = -0.219$ and $\beta = -0.109$, $p < 0.01$), indicating that internet use has a negative effect on patient trust. This outcome is consistent with negative bias theory. To catch the public's attention, information on the internet tends to exaggerate descriptions and concentrate on only one side of medical incidents. Considering the sharp drop in patient trust and severe workplace violence against healthcare providers in China in recent years, the key role of the change in the level of patient trust is clear. The media tends to overemphasize the responsibility of doctors and hospitals, such as a lack of professional medical knowledge, falling morals and harsh medical environments. Moreover, there is little information about the social cost of training a qualified doctor, and how busy doctors are. The media fails to guide patients to fully understand doctors' situation. Negative reports about doctors further affect patients' impressions. Many patients who hold suspect views about physicians may consider health care providers to be "living in their imagination" instead of personally experiencing the real world.

**Table 3.** Benchmark estimates for internet use and patient trust.

| Variables | OLS | | | | Ordered Probit | | | |
|---|---|---|---|---|---|---|---|---|
| | **(1)** | **(2)** | **(3)** | **95% CI** | **(4)** | **(5)** | **(6)** | **95% CI** |
| Internet use | −0.278 *** | −0.245 *** | −0.219 *** | [−0.292, | −0.149 *** | −0.121 *** | −0.109 *** | [−0.144, |
| | (0.028) | (0.037) | (0.036) | −0.145] | (0.013) | (0.018) | (−0.018) | −0.074] |
| Urban | | −0.290 *** | −0.275 *** | [−0.333, | | −0.142 *** | −0.133 *** | [−0.161, |
| | | (0.029) | (0.030) | −0.215] | | (0.014) | (0.014) | −0.104] |
| Gender | | −0.231 *** | −0.239 *** | [−0.291, | | −0.109 *** | −0.114 *** | [−0.139, |
| | | (0.027) | (0.027) | −0.185] | | (0.013) | (0.013) | −0.088] |
| Age | | −0.005 *** | −0.003 *** | [−0.005, | | −0.002 *** | −0.001 ** | [−0.002, |
| | | (0.001) | (0.001) | 0.001] | | (0.001) | (0.001) | −0.001] |
| Education | | −0.007 | 0.002 | [−0.034, | | −0.013 | −0.009 | [−0.026, |
| | | (0.018) | (0.018) | 0.038] | | (0.009) | (0.009) | 0.008] |
| Marital status | | −0.206 *** | −0.216 *** | [−0.290, | | −0.096 *** | −0.102 *** | [−0.137, |
| | | (0.038) | (0.038) | −0.142] | | (0.018) | (0.018) | −0.066] |
| Income | | −0.029 * | −0.035 ** | [−0.067, | | −0.019 ** | −0.020 *** | [−0.036, |
| | | (0.016) | (0.016) | −0.003] | | (0.008) | (0.008) | −0.004] |
| Insurance | | 0.216 *** | 0.177 *** | [0.070, | | 0.094 *** | 0.077 *** | [0.025, |
| | | (0.054) | (0.051) | 0.282] | | (0.026) | (0.025) | 0.127] |
| Trust intention | | 0.535 *** | 0.521 *** | [0.467, | | 0.248 *** | 0.243 *** | [0.216, |
| | | (0.028) | (0.027) | 0.575] | | (0.013) | (0.013) | 0.268] |
| Life satisfaction | | 0.388 *** | 0.384 *** | [0.353, | | 0.191 *** | 0.190 *** | [0.174, |
| | | (0.016) | (0.014) | 0.415] | | (0.008) | (0.007) | 0.204] |
| Health condition | | −0.056 *** | −0.055 *** | [−0.057, | | −0.133 *** | −0.027 *** | [−0.113, |
| | | (0.011) | (0.011) | −0.026] | | (0.016) | (0.005) | −0.002] |
| Province | | | YES | | | | YES | |
| Constant | 6.931 *** | 5.861 *** | 5.972 *** | | | | | |
| | (0.021) | (0.189) | (0.248) | | | | | |
| $R^2$ | 0.004 | 0.067 | 0.104 | | 0.003# | 0.066# | 0.118# | |
| N | 26113 | 26113 | 26113 | | 26113 | 26113 | 26113 | |

Note: * $p < 0.10$, ** $p < 0.05$, *** $p < 0.01$. Robust standard deviations are shown in parentheses. #$R^2$ are Pseudo R2. CI is confidence intervals.

Among covariates, urban and higher-income residents exert a negative and significant impact on patient trust. Coinciding with previous studies [51], urban and high-income residents are usually from higher socioeconomic levels, thus having higher expectations of the medical system and doctors' professional competency. When they are facing negative news and their expectations are broken, patient trust will drop sharply. The negative coefficients of the gender variable indicate that male residents show a lower patient trust compared with female residents. This finding is in line with a previous finding that female breast cancer patients are reported to have a higher level of mean trust [36]. Age tends to be an important factor that affects patient trust. Older people are more likely to be less familiar with internet skills, and have less access to information sources, which means their recognition of physicians is more easily changed. Education seems unrelated to patient trust. There may be two reasons for this: on the one hand, higher education means that residents have a greater knowledge and ability to form a correct understanding of the disease on their own, maintaining trust in doctors. On the other hand, higher-educated people are more likely to keep their individual rationality, and thus show less subjective bias towards physicians. The negative and significant coefficient of marital status indicates married residents hold lower patient trust than unmarried residents. Since divorce experience may hurt trust level, the association between marital status and trust remains confusing, and to be discerned.

Our results show that own medical insurance, self-rated life satisfaction, general trust intention of residents and health conditions are associated positively and significantly with patient trust. These results are consistent with previous studies [52,53], which reported residents with higher self-rated happiness are more likely to trust physicians. People who have bought medical insurance are more likely to hold a higher patient trust. Since

these residents are often more concerned about their health conditions and are willing to acquire health information actively, patient trust can be built from medical knowledge, compared with non-medical insurance holders. Life satisfaction is positively and significantly associated with patient trust, suggesting that residents who are more pleased with life demonstrate higher trust in physicians. Residents who are satisfied with their lives tend to be optimistic about everything and show a higher level of patient trust [54]. A higher general trust intention does benefit patient trust. Patients with better health tend to trust doctors more, which is consistent with Zhang's research results [53].

*4.3. Estimating the Impact of Internet Use on Patient Trust and Its Average Treatment Effect (ATT)*

Although the benchmark model used in this paper controls a series of variables that may affect patient trust, there may still be endogenous problems. Therefore, this study further uses the ESR model for estimation. The results of the ESR models are presented Table 4. As expected, after controlling for individual trust intention, the instrumental variable has a statistically significant impact on internet use, and the results of the outcome equations are also presented in the tables. Since the objective of this study is to estimate the impact of internet use on patient trust based on the variables in Table 3, a detailed explanation of other factors that affect any of the outcome variables is left out, but this information can be provided upon request.

**Table 4.** The determinants of internet use and the determinants of patient trust.

| Variables | First Stage | Patient Trust | |
|---|---|---|---|
| | Internet Usage | Users | Non Users |
| Urban | 0.240 *** | −0.216 *** | −0.283 *** |
| | (10.815) | (−5.400) | (−5.840) |
| Gender | 0.040 * | −0.266 *** | −0.158 *** |
| | (1.930) | (−7.566) | (−3.679) |
| Age | −0.056 *** | −0.018 *** | −0.002 |
| | (−63.973) | (−7.353) | (−0.390) |
| Education | 0.450 *** | 0.095 *** | 0.017 |
| | (33.926) | (3.220) | (0.399) |
| Marital status | −0.058 * | −0.116 ** | −0.191 *** |
| | (−1.746) | (−2.383) | (−2.891) |
| Income | 0.253 *** | −0.029 | 0.011 |
| | (19.791) | (−1.163) | (0.432) |
| Insurance | 0.177 *** | 0.248 *** | 0.147 * |
| | (4.416) | (3.806) | (1.795) |
| Trust intention | 0.052 ** | 0.573 *** | 0.455 *** |
| | (2.547) | (15.902) | (10.886) |
| Life satisfaction | −0.038 *** | 0.365 *** | 0.386 *** |
| | (−3.541) | (18.582) | (18.227) |
| Health condition | 0.093 ** | 0.671 *** | 0.556 *** |
| | (2.547) | (15.902) | (10.886) |
| Constant | −2.828 *** | 5.924 *** | 5.656 *** |
| | (−9.393) | (14.392) | (7.915) |
| Province | YES | YES | YES |
| Mobile Ownership | 1.733 *** | | |
| | (14.661) | | |
| $\rho_1$ | 0.128 *** | | |
| | (0.035) | | |
| $\rho_2$ | 0.134 | | |
| | (0.682) | | |

Note: * $p < 0.10$, ** $p < 0.05$, *** $p < 0.01$. T-value are shown in parentheses.

As depicted in Table 4, $\rho_1$ shows significance, which suggests that the decision to use the internet is not random and that selection bias exists, indicating that the ESR model is appropriate for estimation. Furthermore, the Wald tests for the joint independence of the

two equations for patient trust have a significant sign at the 1% level, which indicates that we can reject the null correlation between the treatment error and the outcome errors. As shown in Table 5, internet use decreases patient trust by nearly 7%. This may not be a large value, but it is important for those patients who find it difficult to trust their physicians: the difference is sufficient to cause a patient to transform from trust to failure to believe what physicians say and do.

**Table 5.** The impact of internet use on patient trust. Average treatment effect on the treated group (ATT).

| Patient Trust | | ATT | T-Value | Change (%) |
|---|---|---|---|---|
| Users | Non-Users | | | |
| 6.655 | 7.154 | −0.499 | −74.947 ** | 6.98% |
| (0.545) | (0.561) | (0.355) | | |

Note: (1) * $p < 0.10$, ** $p < 0.05$, *** $p < 0.01$. Robust standard deviations are shown in parentheses.

### 4.4. Relationship between Information Channel Importance Attitude, Internet Behavior and Patient Trust

The analysis above provides preliminary evidence of the effect of internet use on patient trust. However, attitudes toward the internet and online behavior variation can lead to different personal perceptions. Evidence indicates that attitudes toward the internet can affect anxiety, while spending more time online is associated with more negative perceptions [55–58]. Hence, we further investigate the relationships between attitude toward different information channels, online behavior, and patient trust. Information channel importance is measured by "how important is the television/internet/newspapers/magazines/radio for you to gain information?" in the CFPS 2018 questionnaire. The answers used a 5-point Likert-type agreement scale, in which 1 = "strongly unimportant" and 5 = "strongly important". Residents were asked how much time they spent online during their spare time every week. In total, 6552 respondents answered the question "how often do you use the internet for entertainment (such as to watch videos, download music and so on)", with possible answers ranging from "almost every day" to "never".

First, we use statistical analysis to find initial information about the relationships between patient trust, information channel perception and internet use. As shown in Figure 2, respondents regard television, newspapers, magazines or radio as the more important information access channel, and have higher trust in doctors, while there was no such trend for the internet. Conversely, respondents who rated the internet as a "strongly unimportant" source of information had the highest level of trust in physicians, which is similar to the results of benchmark regression and ESR estimation, since the internet is generally considered a very unimportant source of information by non-internet users. Figure 3 shows the scatter plot and fitting curve for hours spent online per week and patient trust. The horizontal axis is the degree to which respondents perceive the importance of different information acquisition channels. It can be observed that a significant negative relationship still exists between them. As Figure 4 shows, patient trust does not appear to be affected by entertainment time spent on the internet. However, whether the results above are reliable requires further verification.

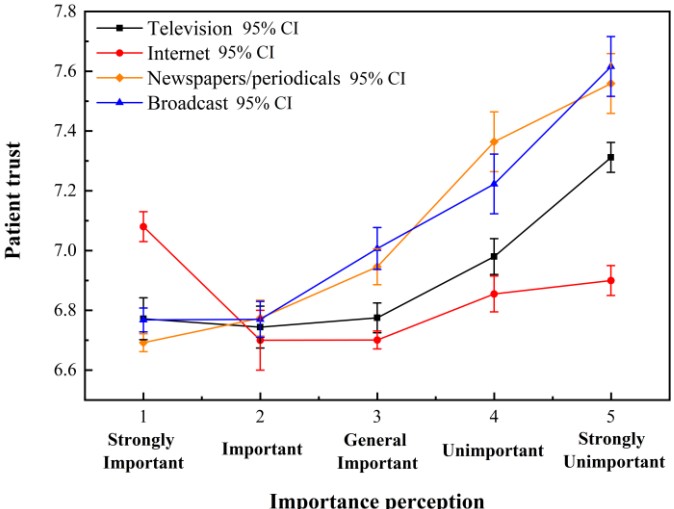

**Figure 2.** The average patient trust in relation to the importance of different information access channels.

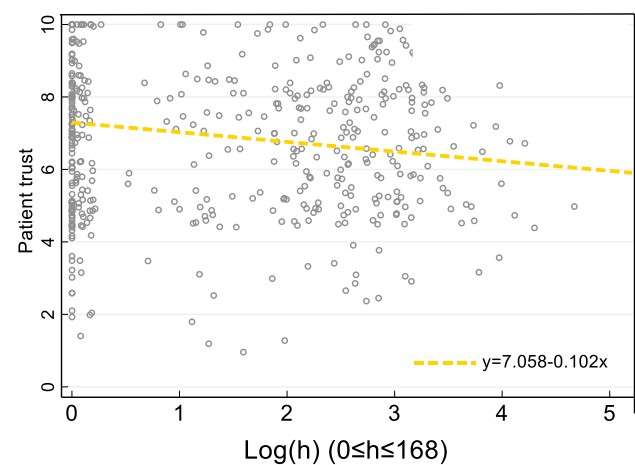

**Figure 3.** Scatter plots and fitting lines for effect of the number of hours spent online per week on patient trust.

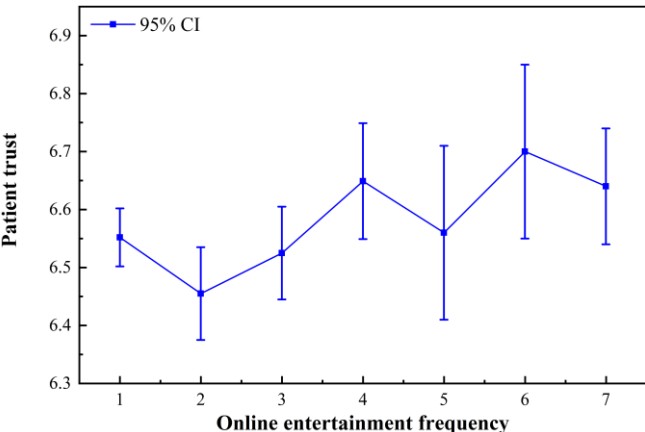

**Figure 4.** Online entertainment frequency's effect on average patient trust.

Table 6 shows the estimated effects on patient trust. We present the results with individual controls and province fixed effects. The coefficient estimates of the information channel in column (1) suggest that people who view the internet as a crucial information

source tend to show lower trust in doctors. This finding is consistent with the previous results obtained. The estimations also reveal that patient trust is improved when television becomes a more essential information channel. According to a survey on media credibility and public trust in China, respondents recognize TV and newspaper as the most dependable information source, and official mouthpieces are perceived to have high reliability [59]. Compared with the internet, the government has greater ability to exert more control on information through traditional media such as television, newspapers, magazines and radio [60]. In authoritarian countries, traditional media are tightly controlled by the government in both the political and health sectors. Traditional media mainly serve propaganda purposes, so they report more positive news and filter out negative news. In the fields of medical and health care, traditional media with a more official background are more likely to spread health knowledge, or report the positive image of doctors, and guide the public to make more positive comments on medical and health departments, hospitals and medical staff; while negative news is more likely to appear in internet media with relatively loose control, new media are more likely to report an event from the perspective of "news" or even criticism. Column (2) shows that spending more time on the internet is indicative of higher trust in physicians, while the estimate for entertainment time in column (3) is not statistically significant. Thus, those who use the internet only for relaxation are less likely to be affected by negative news about care providers.

**Table 6.** Relationships between attitude toward information channels, online behavior and patient trust.

| Variables | OLS | | |
| --- | --- | --- | --- |
| | **(1)** | **(2)** | **(3)** |
| Importance of the internet as information channel | −0.017 ** (0.007) | | |
| Importance of television as information channel | 0.121 *** (0.011) | | |
| Importance of newspapers/magazines as information channel | 0.028 ** (0.013) | | |
| Importance of radio as information channel | 0.077 *** (0.012) | | |
| Time spent on the internet during spare time | | −0.077 *** (0.012) | |
| Entertainment time spent on the internet | | | 0.003 (0.005) |
| Individual characteristics | YES | YES | YES |
| Province | YES | YES | YES |
| N | 25783 | 13702 | 13773 |

Note: (1) * $p < 0.10$, ** $p < 0.05$, *** $p < 0.01$. Robust standard deviations are shown in parentheses.

### 4.5. Subsample Results

We further investigate the heterogeneity of the effect of internet use on patient trust across different demographic groups, focusing on age and education. The results are shown in Figure 5. Although internet use negatively affects patient trust overall, the effect differs in regard to age and education. Specifically, internet use reduces older patient's trust to a greater extent. This implies that older individuals are more likely to be influenced by negative reports and news, and then put less trust in their physicians. Moreover, internet use hurts patient trust only among individuals with lower educational attainment, and who fail to correctly assess information quality on the internet, which leads to much lower patient trust and worse physician relationships. Despite the lack of a significant relationship between high education and patient trust, the coefficient indicates the value of education. Higher educational attainment leads to individuals being more rational, and the emotions of such individuals are less affected by negative news and reports. They can search for reliable information on professional medical websites and improve their knowledge about

their own diseases. When such individuals go to a doctor, they communicate with the care providers actively and improve the doctor–patient relationship. Although one study has found that younger citizens tend to have lower trust in China, our results indicate that trust in health care providers can be improved under internet use conditions. In contrast to older people, young people with more complete internet skills can obtain diverse information. They can obtain opinions from different perspectives, thus developing a more comprehensive and objective mode of analysis. Middle-aged and elderly individuals often lack practical internet skills and have fewer information sources. These individuals can be misled, thereby reducing their trust in their health care providers.

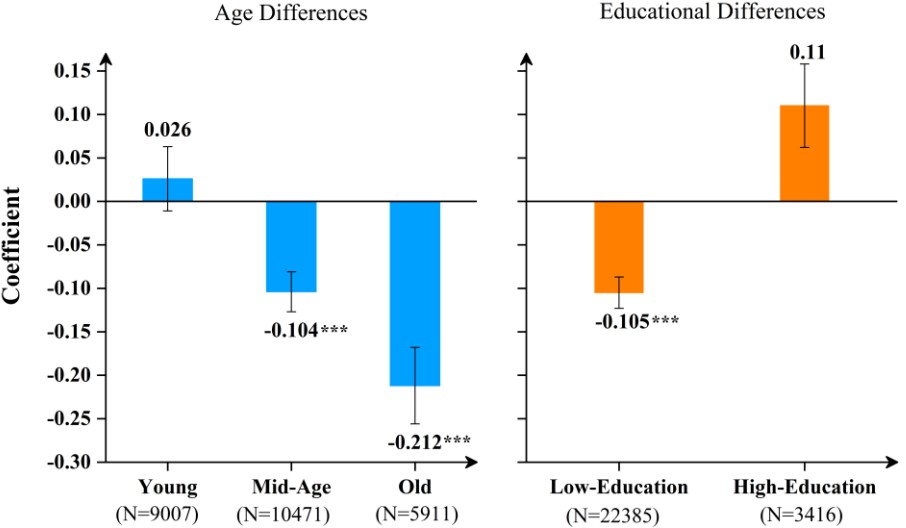

**Figure 5.** Effects of age and educational differences in the impact of internet use on patient trust.

### 4.6. Robustness Check

The goal of propensity score matching is to enable valid comparisons by building a counterfactual control group. Since propensity score matching is widely used to address selection bias, we also apply this method to assess model robustness. In the present study, the regression coefficients of individuals' explanatory variables related to internet choice are estimated; then, the propensity score values of individuals are obtained according to the regression coefficients. We employ three commonly used algorithms: nearest neighbor matching, radius matching, and kernel matching. The results of the ATT estimation are shown in Table 7. Taking nearest neighbor matching as an example, after matching, non-internet users' patient trust is significantly higher than that of internet users. The same conclusion is obtained using the radius matching and kernel matching methods, so the result is robust.

**Table 7.** Propensity score matching results.

| Matching Method | Nearest Neighbor Matching | Radius Matching | Kernel Matching |
| --- | --- | --- | --- |
| Average Effect of Treatment Group | −0.199 ** (0.085) | −0.14 ** (0.065) | −0.15 ** (0.064) |
| T-Stat | −2.33 | −2.14 | −2.34 |
| Individual Characteristic | YES | YES | YES |
| Sample Number of Treatment Group | 13341 | 13341 | 13341 |
| Sample Number of Control Group | 10365 | 10365 | 10365 |

Note: (1) * $p < 0.10$, ** $p < 0.05$, *** $p < 0.01$. Robust standard deviations are shown in parentheses.

Furthermore, we examine patient trust sensitivity to internet use by substituting patient trust into different measurements that reflect respondents' perceptions of the severity of medical problems in China. Table 8 shows the determinants of internet use and medical

problem perception. The similar coefficient indicates the efficiency of the variable measurement: the effect of internet use on physician trust is negative and statistically significant. Moreover, the change in medical problem severity perception estimation is 23.95% (see Table 9), which implies a greater effect of internet usage. Why does this difference exist in similar questions? Feelings of uncertainty may explain this issue. The perception of medical problem severity is closely related to overall medical circumstances, which cannot be decided by respondents themselves, leading to higher uncertainty. However, although patient trust can be affected in various ways, residents often make health care choices according to their social and economic conditions. Patients prefer to choose medical service institutions and doctors that they believe are reliable. In China, people often turn to physicians with whom they have close relationships, thereby leading to higher patient trust. As a result, internet use has a negative effect on patient trust, although this effect is weak compared with the perception of medical problems in China.

**Table 8.** The determinants of internet use and the determinants of medical problem perception.

| Variables | First Stage | Medical Problem Perception | |
|---|---|---|---|
| | Internet Use | Users | Non Users |
| Urban | 0.238 *** | 0.102 ** | 0.003 |
| | (10.729) | (2.319) | (0.063) |
| Gender | 0.034 * | −0.011 | −0.052 |
| | (1.664) | (−0.299) | (−1.107) |
| Age | −0.056 *** | −0.002 | 0.002 |
| | (−63.905) | (−0.780) | (0.436) |
| Education | 0.452 *** | 0.247 *** | −0.062 |
| | (34.090) | (7.378) | (−1.625) |
| Marital Status | −0.056 * | 0.190 *** | −0.137 * |
| | (−1.680) | (3.567) | (−1.906) |
| Income | 0.253 *** | 0.054 ** | −0.029 |
| | (19.869) | (1.977) | (−1.122) |
| Insurance | 0.178 *** | 0.000 | −0.078 |
| | (4.454) | (0.005) | (−0.869) |
| Trust Intention | 0.051 ** | −0.164 *** | −0.169 *** |
| | (2.508) | (−4.155) | (−3.676) |
| Life Satisfaction | −0.038 *** | −0.067 *** | 0.037 |
| | (−3.603) | (−3.113) | (1.578) |
| Health Condition | 0.089 ** | −0.262 *** | −0.249 *** |
| | (2.508) | (−4.155) | (−3.676) |
| Constant | −2.802 *** | 5.848 *** | 5.705 *** |
| | (−9.313) | (12.892) | (7.311) |
| Province | YES | YES | YES |
| Mobile Ownership | 1.707 *** | | |
| | (14.610) | | |
| ρ1 | −0.01 | | |
| | 0.04 | | |
| ρ2 | −0.238 *** | | |
| | 0.05 | | |

Note: (1) * $p < 0.10$, ** $p < 0.05$, *** $p < 0.01$; T-values are shown in parentheses.

**Table 9.** The impact of internet use on medical problem perception. Average treatment effect on the treated group (ATT).

| Medical Problem Perception | | ATT | T-Value | Change (%) |
|---|---|---|---|---|
| Users | Non-Users | | | |
| 7.287 | 5.879 | 1.418 | 285.736 | 23.95% |
| (0.003) | (0.003) | (0.358) | | |

Note: (1) * $p < 0.10$, ** $p < 0.05$, *** $p < 0.01$. Robust standard deviations are shown in parentheses.

## 5. Conclusions and Policy Implications

In China, violence against health care providers remains a severe problem. Lack of trust in physicians is not only a core element of this problem, but is also a worldwide crisis. Exploring measures to improve patient trust is worthwhile for governments to ensure harmonious relationships between patients and doctors. Despite the high internet penetration rate in China, the causal relationship between internet use and patient trust is not fully understood. This paper employs the OLS model and the ordered probit model to explore the impact of internet use on patient trust with data from China Family Panel Studies for 2018.

The results from the benchmark equations reveal that internet use significantly reduces residents' trust in physicians when controlling for resident personal characteristics and province fixed effects. Furthermore, we investigate the effect of information channel importance and online behavior on patients' trust in physicians. The results demonstrate that the more people rely on information from traditional media such as television, newspapers, magazines and radio, the higher their trust in doctors is. In addition, people who spend more time on the internet show lower trust. To account for selection bias, an ESR model is estimated, and the estimation reveals both the actual and counterfactual scenarios. Our results indicate that for internet users, patient trust is approximately 7% lower, which is crucial against China's decreasing patient trust background. The empirical evidence is robust: exploiting PSM with different algorithms and substituting patient trust into medical problem perception yields a negative and significant impact on patient trust. Finally, subsample discussions focusing on education and age are conducted to assess the heterogeneity of internet use. Internet use and patient trust have a significantly negative relationship for residents that are middle-aged or old, and those with low educational attainment, which further verifies the reliability of the main conclusions. Although the treatment effects on young people and highly educated people are not significant, this information provides an important message, and has implications for the government as they seek to manage patient trust effectively.

Policymakers must consider the internet's potential negative impact on patient trust, and promote the relationship between patients and physicians. A small increase in patient trust may help residents transform from suspicious to trusting, and bring benefits to both patients and physicians. Therefore, the government should implement appropriate measures via both the information provider and the information receiver simultaneously. With the development of information and communication technology, the usage scenarios and frequency of internet use will continue to increase in digital China in the future. While realizing that the internet does help to lower information barriers and to bridge the information gap, policymakers should also be alert to the potential risk of the diversified information on the internet. In the construction of the internet, the government should focus not only on information quantity, but also on quality, which is even more crucial. Negative news about the fragile relationship between doctors and patients could be used to gain public attention. The government should improve internet conditions by strengthening the control of internet media, and posting official knowledge and information in a timely manner. Official mouthpieces can provide more positive news and guide citizens' perceptions of medical problems. For information receivers, precise policy implementation is called for because of the complex conditions of those who use the internet in China. Medical knowledge lectures and other activities should be held to improve the level of basic public medical knowledge. Moreover, less educated people and the elderly tend to be more easily lost in the quantity of information on the Internet, and become bad-tempered and have less intention to trust others. Community services, such as training programs that concentrate on rationally using the internet and identifying information quality, also require further consideration.

**Author Contributions:** Conceptualization, L.M. and P.L.; methodology, X.Y.; validation, X.Y. and P.L.; formal analysis, P.L.; investigation, C.H.; resources, L.M.; data curation, L.M.; writing—original draft preparation, L.M.; writing—review and editing, P.L.; supervision, C.H.; project administration, L.M.; funding acquisition, P.L. All authors have read and agreed to the published version of the manuscript.

**Funding:** This research was funded by the National Natural Science Foundation of China, grant number 71974122, 71972127, 71874123, 71503185, and the Shanghai Science and Technology Committee, grant number 19DZ1209202, 21010501800, 22010501900.

**Institutional Review Board Statement:** Not Applicable.

**Informed Consent Statement:** Not applicable.

**Data Availability Statement:** Not applicable.

**Conflicts of Interest:** The authors declare no conflict of interest.

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
