# Peer review of "Does Internet Use Aggravate Public Distrust of Doctors? Evidence from China"

_sustainability, doi:10.3390/su14073959_

Round 1

Reviewer 1 Report

This work evaluates influences of internet on aggression and distrust surrounding doctors. The core measures of the work are drawn from public data.

Overall I think the work is reasonably structured and considered, however, there're many opportunities to refine the results and I have a fundamental concerns about the validity of the results given the binary 'internet use' variable, which appears to dramatically simplify a complex notion, arguably increasing variance and potentially undermining validity. I appreciate the authors efforts to navigate the simplicity of this variable with statistical techniques, though I think another option worth considering would be to perform population checks on this variable, so as to estimate biases in this measure in the current data.

Below are several other points that I encourage the authors to consider in updating the results section: 

Typo in Fig 1 caption, p11. "unimportance" should probably be "unimportant"

Fig 2 is log hours, but not labeled as such. Instead of saying this in the caption show it in the figure and model. Similarly figure 1 would benefit from being labeled with importance levels instead of numeric factors.

Figures should include confidence intervals, especially 1 and 3. Further, many of the tables might make more sense as plots, for example the age and education level results. 

In table 8, is it a typo that the effects are reversed with Radius Matching?

In the future, please make the analysis code available for review. Please also share any pre-registration of this process and analysis.

Reviewer 3 Report

After an exhaustive review of the article, I consider that it is of special relevance and interest given the current situation in which we are immersed. That is why, after reviewing the entire article in its different parts, I consider that the article is suitable for publication. 

The following is the argumentation that I am presenting

The main question addressed by the research focuses on the use of the Internet and the impact this has on users' mistrust of physicians.

I consider the premise pursued by the research to be of special relevance since it highlights an evidence that increases with the increase of violence suffered by doctors in China, aiming to find the root of this problem in order to reduce it on the basis of the results obtained, an aspect in which the research that concerns us is framed and which provides answers to the different questions that arise around this issue.

It is an original approach because until now it is a premise for which no information had been obtained, so the authors' approach to this problem is of special relevance for the readers.

This research contributes to highlight an emerging problem in the country of China that, in many cases, is unknown.

The article presents adequate grammar and coherence throughout.

The conclusions at the end of the article show the repercussions of an inadequate use of certain information offered by the different social networks, specifically the Internet. Effect that political authorities must face in order to solve this problem.

Author Response

Thank you very much for your thoughtful evaluation and recognition of our paper. Best wishes to you!

Reviewer 4 Report

1- 1- The authors should enlist the assistance of a native English-speaking proof reader, because there are some typos and linguistic mistakes that should be fixed.

2- It is recommended that a flowchart be included in the paper to demonstrate the research approach.

3- A survey of the literature is insufficient. It is recommended that the literature review include a few recent work (2018-2021).

Round 2

Reviewer 1 Report

Overall several improvements, in particular I appreciate the addition of more measure of uncertainty and improved diagrams.

In table 1 some confidence interval are not possible, e.g., for Patient trust toward physicians, and for Mobile ownership status, and they often have inconsistent precision from the mean estimates. 

Figure 4 caption typo: frequence should be frequency

Figure 2 and 4 should account for uncertainty by showing confidence intervals on estimates.

I'm marking as major revision even though the fixes I'm suggesting are quite minor, just because I think they are very important.

Round 3

Reviewer 1 Report

I thank the authors for their continued refinements in this work. I'm willing to accept it at this time. 

This manuscript is a resubmission of an earlier submission. The following is a list of the peer review reports and author responses from that submission.